# WHAT SHOULD NOT BE CONTRASTIVE IN CONTRASTIVE LEARNING

**Tete Xiao**
UC Berkeley

**Xiaolong Wang**
UC San Diego

**Alexei A. Efros**
UC Berkeley

**Trevor Darrell**
UC Berkeley

## ABSTRACT

Recent self-supervised contrastive methods have been able to produce impressive transferable visual representations by learning to be invariant to different data augmentations. However, these methods implicitly assume a particular set of representational invariances (e.g., invariance to color), and can perform poorly when a downstream task violates this assumption (e.g., distinguishing red vs. yellow cars). We introduce a contrastive learning framework which does not require prior knowledge of specific, task-dependent invariances. Our model learns to capture varying and invariant factors for visual representations by constructing separate embedding spaces, each of which is invariant to all but one augmentation. We use a multi-head network with a shared backbone which captures information across each augmentation and alone outperforms all baselines on downstream tasks. We further find that the concatenation of the invariant and varying spaces performs best across all tasks we investigate, including coarse-grained, fine-grained, and few-shot downstream classification tasks, and various data corruptions.

## 1 INTRODUCTION

Self-supervised learning, which uses raw image data and/or available pretext tasks as its own supervision, has become increasingly popular as the inability of supervised models to generalize beyond their training data has become apparent. Different pretext tasks have been proposed with different transformations, such as spatial patch prediction (Doersch et al., 2015; Noroozi & Favaro, 2016), colorization (Zhang et al., 2016; Larsson et al., 2016; Zhang et al., 2017), rotation (Gidaris et al., 2018). Whereas pretext tasks aim to recover the transformations between different "views" of the same data, more recent contrastive learning methods (Wu et al., 2018; Tian et al., 2019; He et al., 2020; Chen et al., 2020a) instead try to learn to be *invariant* to these transformations, while remaining discriminative with respect to other data points. Here, the transformations are generated using classic data augmentation techniques which correspond to common pretext tasks, e.g., randomizing color, texture, orientation and cropping.

Yet, the inductive bias introduced through such augmentations is a double-edged sword, as each augmentation encourages invariance to a transformation which can be beneficial in some cases and harmful in others: e.g., adding rotation may help with view-independent aerial image recognition, but significantly downgrade the capacity of a network to solve tasks such as detecting which way is up in a photograph for a display application. Current self-supervised contrastive learning methods assume implicit knowledge of downstream task invariances. In this work, we propose to learn visual representations which capture individual factors of variation in a contrastive learning framework without presuming prior knowledge of downstream invariances.

Instead of mapping an image into a single embedding space which is invariant to all the hand-crafted augmentations, our model learns to construct separate embedding sub-spaces, each of which is sensitive to a specific augmentation while invariant to other augmentations. We achieve this by optimizing multiple augmentation-sensitive contrastive objectives using a multi-head architecture with a shared backbone. Our model aims to preserve information with regard to each augmentation in a unified representation, as well as learn invariances to them. The general representation trained with these augmentations can then be applied to different downstream tasks, where each task is free to selectively utilize different factors of variation in our representation. We consider transfer of either the shared backbone representation, or the concatenation of all the task-specific heads; both

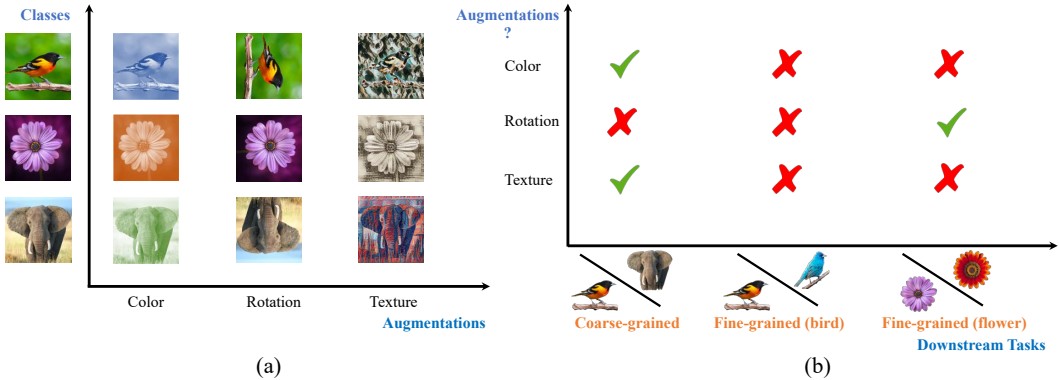

Figure 1: Self-supervised contrastive learning relies on data augmentations as depicted in (a) to learn visual representations. However, current methods introduce inductive bias by encouraging neural networks to be less sensitive to information w.r.t. augmentation, which may help or may hurt. As illustrated in (b), rotation invariant embeddings can help on certain flower categories, but may hurt animal recognition performance; conversely color invariance generally seems to help coarse grained animal classification, but can hurt many flower categories and bird categories. Our method, shown in the following figure, overcomes this limitation.

outperform all baselines; the former uses same embedding dimensions as typical baselines, while the latter provides greatest overall performance in our experiments. In this paper, we experiment with three types of augmentations: rotation, color jittering, and texture randomization, as visualized in Figure 1. We evaluate our approach across a variety of diverse tasks including large-scale classification (Deng et al., 2009), fine-grained classification (Wah et al., 2011; Van Horn et al., 2018), few-shot classification (Nilsback & Zisserman, 2008), and classification on corrupted data (Barbu et al., 2019; Hendrycks & Dietterich, 2019). Our representation shows consistent performance gains with increasing number of augmentations. Our method does not require hand-selection of data augmentation strategies, and achieves better performance against state-of-the-art MoCo baseline (He et al., 2020; Chen et al., 2020b), and demonstrates superior transferability, generalizability and robustness across tasks and categories. Specifically, we obtain around 10% improvement over MoCo in classification when applied on the iNaturalist (Van Horn et al., 2018) dataset.

## 2 BACKGROUND: CONTRASTIVE LEARNING FRAMEWORK

Contrastive learning learns a representation by maximizing similarity and dissimilarity over data samples which are organized into similar and dissimilar pairs, respectively. It can be formulated as a dictionary look-up problem (He et al., 2020), where a given reference image $\mathcal{I}$ is augmented into two views, query and key, and the query token $q$ should match its designated key $k^+$ over a set of sampled negative keys $\{k^-\}$ from other images. In general, the framework can be summarized as the following components: (i) A data augmentation module $\mathcal{T}$ constituting $n$ atomic augmentation operators, such as random cropping, color jittering, and random flipping. We denote a pre-defined atomic augmentation as random variable $X_i$. Each time the atomic augmentation is executed by sampling a specific augmentation parameter from the random variable, i.e., $x_i \sim X_i$. One sampled data augmentation module transforms image $\mathcal{I}$ into a random view $\widetilde{\mathcal{I}}$, denoted as $\widetilde{\mathcal{I}} = \mathcal{T}[x_1, x_2, \ldots, x_n](\mathcal{I})$. Positive pair $(q, k^+)$ is generated by applying two randomly sampled data augmentation on the same reference image. (ii) An encoder network $f$ which extracts the feature $\boldsymbol{v}$ of an image $\mathcal{I}$ by mapping it into a $d$-dimensional space $\mathbb{R}^d$. (iii) A projection head $h$ which further maps extracted representations into a hyper-spherical (normalized) embedding space. This space is subsequently used for a specific pretext task, i.e., contrastive loss objective for a batch of positive/negative pairs. A common choice is InfoNCE (Oord et al., 2018):

$$\mathcal{L}_q = -\log \frac{\exp\left(q{\cdot}k^+/\tau\right)}{\exp\left(q{\cdot}k^+/\tau\right) + \sum_{k^-} \exp\left(q{\cdot}k^-/\tau\right)}, \qquad (1)$$

where $\tau$ is a temperature hyper-parameter scaling the distribution of distances.

As a key towards learning a good feature representation (Chen et al., 2020a), a strong augmentation policy prevents the network from exploiting naïve cues to match the given instances. However, in-

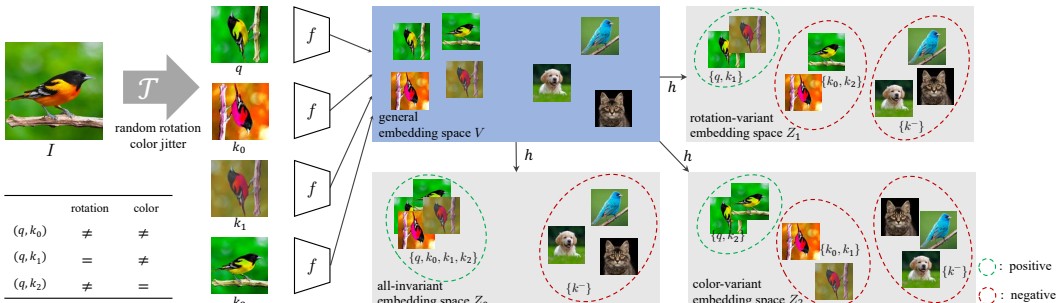

Figure 2: **Framework of the Leave-one-out Contrastive Learning approach**, illustrated with two types of augmentations, i.e., random rotation and color jittering. We generate multiple views with leave-one-out strategy, then project their representations into separate embedding spaces with contrastive objective, where each embedding space is either invariant to all augmentations, or invariant to all but one augmentation. The learnt representation can be the general embedding space $\mathcal{V}$ (blue region), or the concatenation of embedding sub-spaces $\mathcal{Z}$ (grey region). Our results show that either of our proposed representations are able to outperform baseline contrastive embeddings and do not suffer from decreased performance when adding augmentations to which the task is not invariant (i.e., the red X's in Figure 1).

ductive bias is introduced through the selection of augmentations, along with their hyper-parameters defining the strength of each augmentation, manifested in Equation 1 that *any* views by the stochastic augmentation module $\mathcal{T}$ of the same instance are mapped onto the same point in the embedding space. The property negatively affects the learnt representations: 1) Generalizability and transferability are harmed if they are applied to the tasks where the discarded information is essential, e.g., color plays an important role in fine-grained classification of birds; 2) Adding an extra augmentation is complicated as the new operator may be helpful to certain classes while harmful to others, e.g., a rotated flower could be very similar to the original one, whereas it does not hold for a rotated car; 3) The hyper-parameters which control the strength of augmentations need to be carefully tuned for each augmentation to strike a delicate balance between leaving a short-cut open and completely invalidate one source of information.

## 3 LooC: LEAVE-ONE-OUT CONTRASTIVE LEARNING

We propose Leave-one-out Contrastive Learning (LooC), a framework for multi-augmentation contrastive learning. Our framework can selectively prevent information loss incurred by an augmentation. Rather than projecting every view into a single embedding space which is invariant to all augmentations, in our LooC method the representations of input images are projected into several embedding spaces, each of which is *not* invariant to a certain augmentation while remaining invariant to others, as illustrated in Figure 2. In this way, each embedding sub-space is specialized to a single augmentation, and the shared layers will contain both augmentation-varying and invariant information. We learn a shared representation jointly with the several embedding spaces; we transfer either the shared representation alone, or the concatenation of all spaces, to downstream tasks.

**View Generation.** Given a reference image and $n$ atomic augmentations, we first augment the reference image with two sets of independently sampled augmentation parameters into the query view $\mathcal{I}_q$ and the first key view $\mathcal{I}_{k_0}$, i.e., $\mathcal{I}_{\{q,k_0\}} = \mathcal{T}[x_1^{\{q,k_0\}}, x_2^{\{q,k_0\}}, \ldots, x_n^{\{q,k_0\}}](\mathcal{I})$. Additionally, we generate $n$ views from the reference image as extra key views, denoted as $\mathcal{I}_{k_i}, \forall i \in \{1, \ldots, n\}$. For the $i^{th}$ additional key view, the parameter of $i^{th}$ atomic augmentation is copied from it of the query view, i.e., $x_i^{k_i} \equiv x_i^q, \forall i \in \{1, \ldots, n\}$; whereas the parameter of other atomic augmentations are still independently sampled, i.e., $x_j^{k_i} \sim X_j, \forall j \neq i$. For instance, assume that we have a set of two atomic augmentations $\{\texttt{random\_rotation}, \texttt{color\_jitter}\}$, $\mathcal{I}_q$ and $\mathcal{I}_{k_1}$ are always augmented by the same rotation angle but different color jittering; $\mathcal{I}_q$ and $\mathcal{I}_{k_2}$ are always augmented by the same color jittering but different rotation angle; $\mathcal{I}_q$ and $\mathcal{I}_{k_0}$ are augmented independently, as illustrated in the left part of Figure 2.

**Contrastive Embedding Space.** The augmented views are encoded by a neural network encoder $f(\cdot)$ into feature vectors $\boldsymbol{v}^q, \boldsymbol{v}^{k_0}, \cdots, \boldsymbol{v}^{k_n}$ in a joint embedding space $\mathcal{V} \in \mathbb{R}^d$. Subsequently, they are projected into $n+1$ normalized embedding spaces $\mathcal{Z}_0, \mathcal{Z}_1, \cdots, \mathcal{Z}_n \in \mathbb{R}^{d'}$ by projection heads $h : \mathcal{V} \mapsto \mathcal{Z}$, among which $\mathcal{Z}_0$ is invariant to all types of augmentations, whereas $\mathcal{Z}_i$ ($\forall i \in \{1, 2, \cdots, n\}$) is dependent on the $i^{th}$ type of augmentation but invariant to other types of augmentations. In other words, in $\mathcal{Z}_0$ all features $\boldsymbol{v}$ should be mapped to a single point, whereas in $\mathcal{Z}_i$ ($\forall i \in \{1, 2, \cdots, n\}$) only $\boldsymbol{v}^q$ and $\boldsymbol{v}^{k_i}$ should be mapped to a single point while $\boldsymbol{v}^{k_j} \; \forall j \neq i$ should be mapped to $n-1$ separate points, as only $\mathcal{I}_q$ and $\mathcal{I}_{k_i}$ share the same $i^{th}$ augmentation.

We perform contrastive learning in all normalized embedding spaces based on Equation 1, as shown in the right part of Figure 2. For each query $\boldsymbol{z}^q$, denote $\boldsymbol{z}^{k^+}$ as the keys from the *same* instance, and $\boldsymbol{z}^{k^-}$ as the keys from *other* instances. Since all views should be mapped to the single point in $\mathcal{Z}_0$, the positive pair for the query $\boldsymbol{z}_0^q$ is $\boldsymbol{z}_0^{k_0^+}$, and the negative pairs are embeddings of other instances in this embedding space $\{\boldsymbol{z}_0^{k_0^-}\}$; for embedding spaces $\mathcal{Z}_1, \cdots, \mathcal{Z}_n$, the positive pair for the query $\boldsymbol{z}_i^q$ is $\boldsymbol{z}_i^{k_i^+}$, while the negative pairs are embeddings of other instances in this embedding space $\{\boldsymbol{z}_i^{k_i^-}\}$, *and* $\{\boldsymbol{z}_i^{k_j^+} \mid \forall j \in \{0, 1, \cdots, n\} \text{ and } j \neq i\}$, which are the embeddings of the *same* instance with different $i^{th}$ augmentation. The network then learns to be sensitive to one type of augmentation while insensitive to other types of augmentations in one embedding space. Denote $E_{i,j}^{\{+,-\}} = \exp(\boldsymbol{z}_i^q \cdot \boldsymbol{z}_i^{k_j^{\{+,-\}}} / \tau)$. The overall training objective for $q$ is:

$$\mathcal{L}_q = -\frac{1}{n+1}\left(\log \frac{E_{0,0}^+}{E_{0,0}^+ + \sum_{k^-} E_{0,0}^-} + \sum_{i=1}^n \log \frac{E_{i,i}^+}{\sum_{j=0}^n E_{i,j}^+ + \sum_{k^-} E_{i,i}^-}\right), \tag{2}$$

The network must preserve information w.r.t. all augmentations in the general embedding space $\mathcal{V}$ in order to optimize the combined learning objectives of all normalized embedding spaces.

**Learnt representations.** The representation for downstream tasks can be from the general embedding space $\mathcal{V}$ (Figure 2, blue region), or the concatenation of all embedding sub-spaces (Figure 2, grey region). LooC method returns $\mathcal{V}$; we term the implementation using the concatenation of all embedding sub-spaces as LooC++.

## 4 EXPERIMENTS

**Methods.** We adopt Momentum Contrastive Learning (MoCo) (He et al., 2020) as the backbone of our framework for its efficacy and efficiency, and incorporate the improved version from (Chen et al., 2020b). We use three types of augmentations as pretext tasks for static image data, namely color jittering (including random gray scale), random rotation ($90°$, $180°$, or $270°$), and texture randomization (Gatys et al., 2016; Geirhos et al., 2018) (details in the Appendix). We apply random-resized cropping, horizontal flipping and Gaussian blur as augmentations without designated embedding spaces. Note that random rotation and texture randomization are not utilized in state-of-the-art contrastive learning based methods (Chen et al., 2020a; He et al., 2020; Chen et al., 2020b) and for good reason, as we will empirically show that naïvely taking these augmentations negatively affects the performance on some specific benchmarks. For LooC++, we include `Conv5` block into the projection head $h$, and use the concatenated features at the last layer of `Conv5`, instead of the last layer of $h$, from each head. Note than for both LooC and LooC++ the augmented additional keys are only fed into the key encoding network, which is not back-propagated, thus it does not much increase computation or GPU memory consumption.

**Datasets and evaluation metrics.** We train our model on the 100-category ImageNet (IN-100) dataset, a subset of the ImageNet (Deng et al., 2009) dataset, for fast ablation studies of the proposed framework. We split the subset following (Tian et al., 2019). The subset contains $\sim$125k images, sufficiently large to conduct experiments of statistical significance. After training, we adopt *linear* classification protocol by training a *supervised* linear classifier on *frozen* features of feature space $\mathcal{V}$ for LooC, or concatenated feature spaces $\mathcal{Z}$ for LooC++. This allows us to directly verify the quality of features from a variation of models, yielding more interpretable results. We test the models on various downstream datasets (more information included in the Appendix): 1) IN-100 validation set; 2) The iNaturalist 2019 (iNat-1k) dataset (Van Horn et al., 2018), a large-scale classification dataset

Table 1: **Classification accuracy on 4-class rotation and IN-100** under linear evaluation protocol. Adding rotation augmentation into baseline MoCo significantly reduces its capacity to classify rotation angles while downgrades its performance on IN-100. In contrast, our method better leverages the information gain of the new augmentation.

| model | Rotation Acc. | IN-100 top-1 | IN-100 top-5 |
|---|---|---|---|
| Supervised | 72.3 | 83.7 | 95.7 |
| MoCo | 61.1 | 81.0 | 95.2 |
| MoCo + Rotation | 43.3 | 79.4 | 94.1 |
| MoCo + Rotation (same for $q$ and $k$) | 45.5 | 78.1 | 94.3 |
| LooC + Rotation [ours] | 65.2 | 80.2 | 95.5 |

Table 2: **Evaluation on multiple downstream tasks**. Our method demonstrates superior generalizability and transferability with increasing number of augmentations.

| model | Augmentation Color | Augmentation Rotation | iNat-1k top-1 | iNat-1k top-5 | CUB-200 top-1 | CUB-200 top-5 | Flowers-102 5-shot | Flowers-102 10-shot | IN-100 top-1 | IN-100 top-5 |
|---|---|---|---|---|---|---|---|---|---|---|
| MoCo | ✓ | | 36.2 | 62.0 | 36.7 | 64.7 | 67.9 ($\pm$ 0.5) | 77.3 ($\pm$ 0.1) | 81.0 | 95.2 |
| LooC | ✓ | | 41.2 | 67.0 | 40.1 | 69.7 | 68.2 ($\pm$ 0.6) | 77.6 ($\pm$ 0.1) | 81.1 | 95.3 |
| | | ✓ | 40.0 | 65.4 | 38.8 | 67.0 | 70.1 ($\pm$ 0.4) | 79.3 ($\pm$ 0.1) | 80.2 | 95.5 |
| | ✓ | ✓ | 44.0 | 69.3 | 39.6 | 69.2 | 70.9 ($\pm$ 0.3) | 80.8 ($\pm$ 0.2) | 79.2 | 94.7 |
| LooC++ | ✓ | ✓ | 46.1 | 71.5 | 39.3 | 69.3 | 68.1 ($\pm$ 0.4) | 78.8 ($\pm$ 0.2) | 81.2 | 95.2 |

containing 1,010 species. Top-1 and top-5 accuracy on this dataset are reported; 3) The Caltech-UCSD Birds 2011 (CUB-200) dataset (Wah et al., 2011), a fine-grained classification dataset of 200 bird species. Top-1 and top-5 classification accuracy are reported. 4) VGG Flowers (Flowers-102) dataset (Nilsback & Zisserman, 2008), a consistent of 102 flower categories. We use the dataset for few-shot classification and report 5-shot and 10-shot classification accuracy over 10 trials within 95% confidence interval. Unlike many few-shot classification methods which conduct evaluation on a subset of categories, we use all 102 categories in our study; 5) ObjectNet dataset (Barbu et al., 2019), a test set collected to intentionally show objects from new viewpoints on new backgrounds with different rotations of real-world images. We only use the 13 categories which overlap with IN-100, termed as ON-13; 6) ImageNet-C dataset (Hendrycks & Dietterich, 2019), a benchmark for model robustness of image corruptions. We use the 100 categories as IN-100, termed as IN-C-100. Note that ON and IN-C are test sets, so we do not train a supervised linear classifier exclusively while directly benchmark the linear classifier trained on IN-100 instead.

**Implementation details.** We closely follow (Chen et al., 2020b) for most training hyper-parameters. We use a ResNet-50 (He et al., 2016) as our feature extractor. We use a two-layer MLP head with a 2048-d hidden layer and ReLU for each individual embedding space. We train the network for 500 epochs, and decrease the learning rate at 300 and 400 epochs. We use separate queues (He et al., 2020) for individual embedding space and set the queue size to 16,384. Linear classification evaluation details can be found in the Appendix. The batch size during training of the backbone and the linear layer is set to 256.

**Study on augmentation inductive biases.** We start by designing an experiment which allows us to directly measure how much an augmentation affects a downstream task which is sensitive to the augmentation. For example, consider two tasks which can be defined on IN-100: Task A is 4-category classification of rotation degrees for an input image; Task B is 100-category classification of ImageNet objects. We train a supervised *linear* classifier for task A with randomly rotated IN-100 images, and another classifier for task B with *unrotated* images. In Table 1 we compare the accuracy of the original MoCo (w/o rotation augmentation), MoCo w/ rotation augmentation, and our model w/ rotation augmentation. A priori, with no data labels to perform augmentation selection, we have no way to know if rotation should be utilized or not. Adding rotation into the set of augmentations for MoCo downgrades object classification accuracy on IN-100, and significantly reduces the capacity of the baseline model to distinguish the rotation of an input image. We further implement a variation enforcing the random rotating angle of query and key always being the same. Although it marginally increases rotation accuracy, IN-100 object classification accuracy further drops, which is inline with our hypothesis that the inductive bias of discarding certain type of information introduced by adopting an augmentation into contrastive learning objective is significant and cannot be trivially resolved by tuning the distribution of input images. On the other hand, our method with rotation augmentation not only sustains accuracy on IN-100, but also leverages the information gain

Table 3: **Evaluation on datasets of real-world corruptions.** Rotation augmentation is beneficial for ON-13, and texture augmentation if beneficial for IN-C-100.

| model | Aug. | | ON-13 | | IN-C-100 (top-1) | | | | | | IN-100 | |
|---|---|---|---|---|---|---|---|---|---|---|---|---|
| | Rot. | Tex. | top-1 | top-5 | Noise | Blur | Weather | Digital | All | $d \geq 3$ | top-1 | top-5 |
| Supervised | | | 30.9 | 54.8 | 28.4 | 47.1 | 44.9 | 58.5 | 47.2 | 36.5 | 83.7 | 95.7 |
| MoCo | | | 29.2 | 54.2 | 37.9 | 38.5 | 47.7 | 60.1 | 48.2 | 37.2 | 81.0 | 95.2 |
| LooC | ✓ | | 34.2 | 59.6 | 31.3 | 33.1 | 42.4 | 54.9 | 42.7 | 31.8 | 80.2 | 95.5 |
| | | ✓ | 30.1 | 54.1 | 42.4 | 39.6 | 54.0 | 61.9 | 51.3 | 41.9 | 81.0 | 94.7 |
| | ✓ | ✓ | 33.3 | 59.2 | 37.0 | 35.2 | 50.2 | 56.9 | 46.5 | 37.2 | 79.4 | 94.3 |
| LooC++ | ✓ | ✓ | 32.6 | 57.3 | 38.3 | 37.6 | 52.0 | 60.0 | 48.8 | 38.9 | 82.1 | 95.1 |

Table 4: **Comparisons of LooC vs. MoCo** trained with all augmentations.

| Model | IN-100 | | iNat-1k | | Flowers-102 | | IN-C-100 |
|---|---|---|---|---|---|---|---|
| | top-1 | top-5 | top-1 | top-5 | 5-shot | 10-shot | all-top-1 |
| MoCo | 77.9 | 93.7 | 39.5 | 65.1 | 72.1 ($\pm$ 0.4) | 81.1 ($\pm$ 0.2) | 47.4 |
| LooC | 78.5 | 94.0 | 41.7 | 67.5 | 72.1 ($\pm$ 0.7) | 81.4 ($\pm$ 0.2) | 45.4 |
| MoCo++ | 80.8 | 94.6 | 43.4 | 68.5 | 70.0 ($\pm$ 0.8) | 80.5 ($\pm$ 0.3) | 48.3 |
| LooC++ | 82.2 | 95.3 | 45.9 | 71.4 | 71.0 ($\pm$ 0.7) | 81.9 ($\pm$ 0.3) | 48.0 |

Table 5: **Comparisons of concatenating features from different embedding spaces in LooC++** jointly trained on color, rotation and texture augmentations. Different downstream tasks show non-identical preferences for augmentation-dependent or invariant representations.

| Model | Variance Head | | | IN-100 | | iNat-1k | | Flowers-102 | | IN-C-100 |
|---|---|---|---|---|---|---|---|---|---|---|
| | Col. | Rot. | Tex. | top-1 | top-5 | top-1 | top-5 | 5-shot | 10-shot | all-top-1 |
| LooC++ | | | | 78.5 | 94.3 | 38.5 | 64.7 | 68.6 ($\pm$ 0.6) | 77.6 ($\pm$ 0.1) | 48.0 |
| | ✓ | | | 79.7 | 94.4 | 42.9 | 68.7 | 69.1 ($\pm$ 0.7) | 79.5 ($\pm$ 0.2) | 47.1 |
| | | ✓ | | 81.5 | 94.9 | 41.4 | 67.4 | 70.5 ($\pm$ 0.6) | 80.0 ($\pm$ 0.2) | 52.6 |
| | | | ✓ | 80.3 | 94.9 | 43.0 | 68.6 | 70.4 ($\pm$ 0.5) | 80.5 ($\pm$ 0.2) | 44.1 |
| | ✓ | ✓ | ✓ | 82.2 | 95.3 | 45.9 | 71.4 | 71.0 ($\pm$ 0.7) | 81.9 ($\pm$ 0.3) | 48.0 |

of the new augmentation. We can include all augmentations with our LooC multi-self-supervised method and obtain improved performance across all condition without any downstream labels or a prior knowledged invariance.

**Fine-grained recognition results.** A prominent application of unsupervised learning is to learn features which are transferable and generalizable to a variety of downstream tasks. To fairly evaluate this, we compare our method with original MoCo on a diverse set of downstream tasks. Table 2 lists the results on iNat-1k, CUB-200 and Flowers-102. Although demonstrating marginally superior performance on IN-100, the original MoCo trails our LooC counterpart on all other datasets by a noticeable margin. Specifically, applying LooC on random color jitering boosts the performance of the baseline which adopts the same augmentation. The comparison shows that our method can better preserve color information. Rotation augmentation also boosts the performance on iNat-1k and Flowers-102, while yields smaller improvements on CUB-200, which supports the intuition that some categories benefit from rotation-invariant representations while some do not. The performance is further boosted by using LooC with both augmentations, demonstrating the effectiveness in simultaneously learning the information w.r.t. multiple augmentations.

Interestingly, LooC++ brings back the slight performance drop on IN-100, and yields more gains on iNat-1k, which indicates the benefits of explicit feature fusion without hand-crafting what should or should not be contrastive in the training objective.

**Robustness learning results.** Table 3 compares our method with MoCo and supervised model on ON-13 and IN-C-100, two testing sets for real-world data generalization under a variety of noise conditions. The linear classifier is trained on standard IN-100, without access to the testing distribution. The fully supervised network is most sensitive to perturbations, albeit it has highest accuracy on the source dataset IN-100. We also see that rotation augmentation is beneficial for ON-13, but significantly downgrades the robustness to data corruptions in IN-C-100. Conversely, texture randomization increases the robustness on IN-C-100 across all corruption types, particularly significant on "Blur" and "Weather", and on the severity level above or equal to 3, as the representations must be insensitive to local noise to learn texture-invariant features, but its improvement on ON-13 is marginal. Combining rotation and texture augmentation yields improvements on both datasets, and LooC++ further improves its performance on IN-C-100.

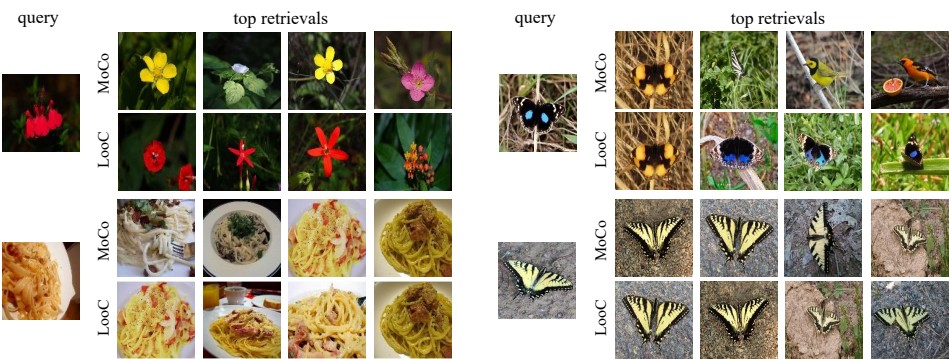

Figure 3: **Top nearest-neighbor retrieval results** of LooC vs. corresponding invariant MoCo baseline with color (left) and rotation (right) augmentations on IN-100 and iNat-1k. The results show that our model can better preserve information dependent on color and rotation despite being trained with those augmentations.

**Qualitative results.**  In Figure 3 we show nearest-neighbor retrieval results using features learnt with LooC vs. corresponding MoCo baseline. The top retrieval results demonstrate that our model can better preserve information which is not invariant to the transformations presented in the augmentations used in contrastive learning.

**Ablation: MoCo w/ all augmentations vs. LooC.**  We compare our method and MoCo trained with all augmentations. We also add multiple `Conv5` heads to MoCo, termed as MoCo++, for a fair comparison with LooC++. The results are listed in Table 4. Using multiple heads boosts the performance of baseline MoCo, nevertheless, our method achieves better or comparable results compared with its baseline counterparts.

Note that the results in Table 2 to 5 should be interpreted in the broader context of Table 1. Table 1 illustrates the catastrophic consequences of not separating the varying and invariant factors of an augmentation (in this case, rotation). It can be imagined that if we add "rotation classification" as one downstream task in Table 4, MoCo++ will perform as poorly as in Table 1. The key of our work is to avoid what has happened in Table 1 and simultaneously boosts performance.

**Ablation: Augmentation-dependent embedding spaces vs. tasks.**  We train a LooC++ with all types of augmentations, and subsequently train multiple linear classifiers with concatenated features from different embedding spaces: all-invariant, color, rotation and texture. Any additional variance features boost the performance on IN-100, iNat-1k and Flowers-102. Adding texture-dependent features decreases the performance on IN-C-100: Textures are (overly) strong cues for ImageNet classification (Geirhos et al., 2018), thus the linear classifier is prone to use texture-dependent features, loosing the gains of texture invariance. Adding rotation-dependent features increases the performance on IN-C-100: Rotated objects of most classes in IN-100 are rare, thus the linear classifier is prone to use rotation-dependent features, so that drops on IN-C-100 triggered by rotation-invariant augmentation are re-gained. Using all types of features yields best performance on IN-100, iNat-1k and Flowers-102; the performance on IN-C-100 with all augmentations remains comparable to MoCo, which does not suffer from loss of robustness introduced by rotation invariance.

In Figure 4 we show the histogram of correct predictions (activations×weights of classifier) by each augmentation-dependent head of a few instances from IN-100 and iNat-1k. The classifier prefers texture-dependent information over other kinds on an overwhelmingly majority of samples from IN-100, even for classes where shape is supposed to be the dominant factor, such as "pickup" and "mixing bowl" ((a), top row). This is consistent with the findings from (Geirhos et al., 2018) that ImageNet-trained CNNs are strongly biased towards texture-like representations. Interestingly, when human or animal faces dominant an image ((a), bottom-left), LooC++ sharply prefers rotation-dependent features, which also holds for face recognition of humans. In contrast, on iNat-1k LooC++ prefers a more diverse set of features, such as color-dependent feature for a dragonfly species, rotation and texture-dependent features for birds, as well as rotation-invariant features for flowers. Averaged over the datasets, the distribution of classifier preferences is more balanced on iNat-1k than IN-100, as can be seen from the entropy that the distribution on iNat-1k is close to 2

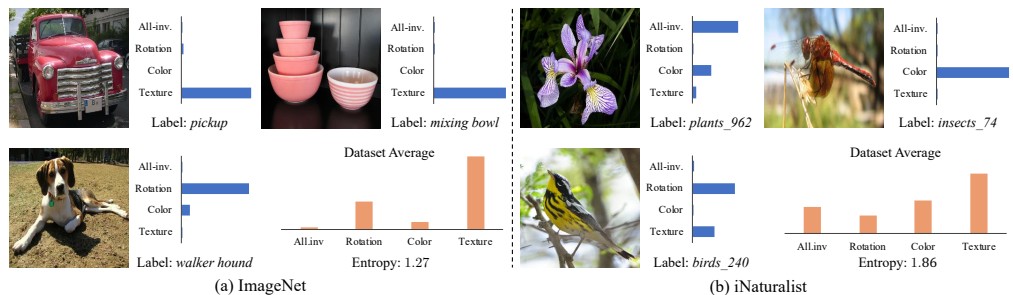

Figure 4: **Histograms of correct predictions (activations×weights of classifier) by each augmentation-dependent head** from IN-100 and iNat-1k. The classifier on IN-100 heavily relies on texture-dependent information, whereas it is much more balanced on iNat-1k. This is consistent with the improvement gains observed when learning with multiple augmentations.

bits, whereas it is close to 1 bit on IN-100, as it is dominated by only two elements. It corroborates the large improvements on iNat-1k gained from multi-dependent features learnt by our method.

## 5 RELATED WORK

**Pretext Tasks.** In computer vision, feature design and engineering used to be a central topic before the wide application of deep learning. Researchers have proposed to utilize cue combination for image retrieval and recognition tasks (Martin et al., 2004; Frome et al., 2007a;b; Malisiewicz & Efros, 2008; Rabinovich et al., 2006). For example, the local brightness, color, and texture features are combined together to represent an image and a simple linear model can be trained to detect boundaries (Martin et al., 2004). Interestingly, the recent development of unsupervised representation learning in deep learning is also progressed by designing different self-supervised pretext tasks (Wang & Gupta, 2015; Doersch et al., 2015; Pathak et al., 2016; Noroozi & Favaro, 2016; Zhang et al., 2016; Gidaris et al., 2018; Owens et al., 2016). For example, relative patch prediction (Doersch et al., 2015) and rotation prediction (Gidaris et al., 2018) are designed to discover the underlined structure of the objects; image colorization task (Zhang et al., 2016) is used to learn representations capturing color information. The inductive bias introduced by each pretext task can often be associated with a corresponding hand-crafted descriptor.

**Multi-Task Self-Supervised Learning.** Multi-task learning has been widely applied in image recognition (Kokkinos, 2017; Teichmann et al., 2018; He et al., 2017). However, jointly optimizing multiple tasks are not always beneficial. As shown in Kokkinos (2017), training with two tasks can yield better performance than seven tasks together, as some tasks might be conflicted with each other. This phenomenon becomes more obvious in multi-task self-supervised learning (Doersch & Zisserman, 2017; Wang et al., 2017; Pinto & Gupta, 2017; Piergiovanni et al., 2020; Alwassel et al., 2019) as the optimization goal for each task can be very different depending on the pretext task. To solve this problem, different weights for different tasks are learned to optimize for the downstream tasks (Piergiovanni et al., 2020). However, searching the weights typically requires labels, and is time-consuming and does not generalize to different tasks. To train general representations, researchers have proposed to utilize sparse regularization to factorize the network representations to encode different information from different tasks (Doersch & Zisserman, 2017; Misra et al., 2016). In this paper, we also proposed to learn representation which can factorize and unify information from different augmentations. Instead of using sparse regularization, we define different contrastive learning objective in a multi-head architecture.

**Contrastive Learning.** Instead of designing different pretext tasks, recent work on contrastive learning (Wu et al., 2018; Oord et al., 2018; Tian et al., 2019; He et al., 2020; Misra & van der Maaten, 2020; Chen et al., 2020a) trained networks to be invariant to various corresponding augmentations. Researchers (Chen et al., 2020a) elaborated different augmentations and pointed out which augmentations are helpful or harmful for ImageNet classification. It is also investigated in Tian et al. (2019) that different augmentations can be beneficial to different downstream tasks. Instead of enumerating all the possible selections of augmentations, we proposed a unified framework which captures different factors of variation introduced by different augmentations.

## 6 CONCLUSIONS

Current contrastive learning approaches rely on specific augmentation-derived transformation invariances to learn a visual representation, and may yield suboptimal performance on downstream tasks if the wrong transformation invariances are presumed. We propose a new model which learns both transformation dependent and invariant representations by constructing multiple embeddings, each of which is *not* contrastive to a single type of transformation. Our framework outperforms baseline contrastive method on coarse-grained, fine-grained, few-shot downstream classification tasks, and demonstrates better robustness of real-world data corruptions.

## ACKNOWLEDGEMENT

Prof. Darrell's group was supported in part by DoD, NSF, BAIR, and BDD. Prof. Wang's group was supported, in part, by gifts from Qualcomm and TuSimple. We would like to thank Allan Jabri, Colorado Reed and Ilija Radosavovic for helpful discussions.

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

## A  AUGMENTATION DETAILS

Following (Chen et al., 2020b), we set the probability of color jittering to 0.8, with (`brightness`, `contrast`, `saturation`, `hue`) as (0.4, 0.4, 0.4, 0.1), and probability of random scale to 0.2. We set the probability of random rotation and texture randomization as 0.5.

## B  DATASETS

**iNat-1k**, a large-scale classification dataset containing 1,010 species with a combined training and validation set of 268,243 images. We randomly reallocate 10% of training images into the validation set as the original validation set is relatively small.

**CUB-200**, which contains 5,994 training and 5,794 testing images of 200 bird species.

**Flowers-102**, which contains 102 flower categories consisting of between 40 and 258 images.

**ObjectNet**, a test set collected to intentionally show objects from new viewpoints on new backgrounds with different rotations of real-world images. It originally has 313-category. We only use the 13 categories which overlap with IN-100.

**ImageNet-C**, which consists of 15 diverse corruption types applied to validation images of ImageNet.

## C  LINEAR CLASSIFICATION

We train the linear layer for 200 epochs for IN-100 and CUB-200, 100 epochs for iNat-1k, optimized by momentum SGD with a learning rate of 30 decreased by 0.1 at 60% and 80% of training schedule; for Flowers-102 we train the linear layer with Adam optimizer for 250 iterations with a learning rate of 0.03.

## D  LEAVE-ONE-OUT VS. ADD-ONE AUGMENTATION

Table 6: Leave-one-out vs. add-one Augmentation. *: Default (none add-one) augmentation strategy.

| model | Augmentation | | IN-100 | |
|-------|------|----------|-------|-------|
| | Color | Rotation | top-1 | top-5 |
| MoCo | ✓ | | 81.0 | 95.2 |
| | ✓ | ✓ | 79.4 | 94.1 |
| MoCo + AddOne | ✓ | | 74.9 | 92.5 |
| | * | ✓ | 79.3 | 94.4 |
| LooC [ours] | ✓ | | 81.1 | 95.3 |
| | * | ✓ | 80.2 | 95.5 |

A straight-forward alternative for our leave-one-out augmentation strategy is add-one augmentation. Instead of applying all augmentations and augmenting two views in the same manner, add-one strategy keeps the query image unaugmentated, while in each augmentation-specific view the designated type of augmentation is applied. The results are shown in Table 6. Add-one strategy oversimplifies the instance discrimination task, e.g., leaving color augmentation out of query view makes it very easy for the network to spot the same instance out of a set of candidates. Our leave-one-out strategy does not suffer such degeneration.

# E  IMAGENET-1K EXPERIMENTS

Table 7: Results of models trained on 1000 category ImageNet and fine-tuned on iNat-1k following linear classification protocol.

| model | iNat-1k | |
|---|---|---|
| | top-1 | top-5 |
| MoCo | 47.8 | 74.3 |
| LooC++ [ours] | 51.2 | 76.5 |

We conduct experiments on 1000 category full ImageNet dataset. The models are trained by self-supervised learning on IN-1k, and fine-tuned on iNat-1k following linear classification protocol. Our model is trained with all augmentations, i.e., color, rotation and texture. Results are reported in Table 7.

# F  DISCUSSIONS

## F.1  THE DIMENSIONS OF MOCO, LOOC, LOOC++

The representations of MoCo and LooC are of exactly the same dimension (2048); same for MoCo++ and LooC++ (2048 * # augmentations). It is specifically designed for fair comparisons.

## F.2  ARE THE HYPER-PARAMETERS TUNED SPECIFICALLY FOR OUR SUBSETS?

No, except that we increase the number of training epochs as the amount of data increases. *We did not specifically tune the baseline so that our method can outperform it most; on the contrary, we first made baseline as strong as possible, then directly applied the same hyper-parameters to our method.* The subset of ImageNet100 behaviors similarly as ImageNet1k; our baseline already significantly outperforms the best method on the same subset from previous literature (75.8% CMC vs. 81.0% top1 [ours]), and since our method is derived from MoCo, they are directly comparable.

