# OpenReview forum: "What Should Not Be Contrastive in Contrastive Learning"
_ICLR.cc/2021/Conference — ICLR 2021 Poster_

### Official Review · AnonReviewer4 · 2020-10-27
**Elegant approach and well-written paper**

**Rating:** 7
**Confidence:** 3

**Review:**

The paper addresses potential information loss in contrastive learning, when the invariance to a variety of augmentations may be suboptimal for other downstream tasks that require the model to learn those augmentations as discriminative features (e.g: colour in fine-grained bird classification).
The approach, LooC, is built on the backbone of MoCo. The representations of images are projected into multiple embedding spaces, each of which is sensitive to an augmentation and invariant to others. Then a shared representation is jointly learned with multiple embedding spaces and then the shared representation or the concatenation of the learned sub-spaces, are fed to downstream tasks.

Strength:
+ Simple, yet elegant approach.
+ Paper is well-written and easy to follow.
+ The approach is computationally efficient, as the additional keys (compared to MoCo) are not back-propagated, and just fed to the encoder network, but it still outperforms the baselines.
+ Results are presented on variant downstream tasks, ranging from fine- and coarse- grained classification, few-shot classification, to improved robustness on natural corruptions qualitative retrieval task.
+ Comprehensive ablation study and experiments.

Comments (these points are not part of my decision assessment.):
+ I am curious how the model can be adapted for the generalized zero shot learning as a downstream task.
+ a very small typo in Experiments: “Note than” to “Note that”
#######################################
After rebuttal:
Thanks for your response.
I read authors response to my question and as well as other reviewers feedback. I will keep my rating as it is.

* no effect on the rating: a point on the question on the *generalized* zero-shot learning as a downstream task, is if employing this approach, improve the performance on unseen classes while is not negatively impacting the performance for seen classes.

---

> ### Author Response · Authors · 2020-11-19
> **Response from the Authors**
>
> We would like to thank the reviewer for the comments. Please find the following for our response.
>
> **Q1. How the model can be adapted for the generalized zero shot learning as a downstream task.**
>
> **A1.** Usually distance over certain metric (cos, L1, etc.) in an embedding space can be used as a classifier for zero-shot learning. The general embedding space in LooC can thus be directly utilized to compute such distance. LooC++ has more flexibility: the sub-spaces can be concatenated for computing distance; certain zero-shot feature fusion techniques can also be used with the multiple sub embedding spaces, such as [1].
>
> [1] Selecting Relevant Features from a Multi-domain Representation for Few-shot Classification, Dvornik et al.

---

### Official Review · AnonReviewer3 · 2020-10-28
**interesting findings regarding augmentation vs embedding subspaces.**

**Rating:** 6
**Confidence:** 3

**Review:**

The current mainstream way of doing contrastive learning in Instance discrimination is to train the network to associate  two independently augmented versions of the same image. The augmentation process often consists of several stages such as cropping, blurring, etc.

The authors argue that some augmentation (such as rotation and texture) are bad when used in a general augmentation pipeline. They propose to use embedding sup-spaces during Instance discrimination learning that would effectively learn from the previously hard-to-learn augmentation. The key is that in each different subspace, the definition of positive/negative samples will change.

The number of augmentation-specific embedding subspaces are correlated with the kinds of augmentation. With 2 or 3 subspaces, the authors can achieve some gain over MoCo (their baseline method) on fine grain datasets. But I wonder if the method can generalize to more subspaces and achieve gain over MoCo on larger datasets like full ImageNet?

Difference between LooC , LooC++, MoCo++. The key difference is that LooC++ is returning the concat of all embedding sub-spaces features. In Table 4, the comparison between MoCo++ and LooC++ seem to suggest there is little difference in overall performance. Other than the performance difference, what is the key difference between MoCo++ and LooC++? I assume both had momentum encoders, separate queues for each subspaces, etc.

typo:
Page 4, penultimate paragraph: “Note that[than] for both LooC and LooC++ ….”

---

> ### Author Response · Authors · 2020-11-19
> **Response from the Authors**
>
> We would like to thank the reviewer for the comments. Please find the following for our response.
>
> **Q1. Generalize to more subspaces and larger datasets.**
>
> **A1.** We’re confident that it can easily generalize to 4 and 5 subspaces, as we have seen from going from 1 to 3. We conjecture that it is less so going to 10 subspaces, as the embedding space will get saturated and hard to train. A straight-forward hypothesis is to combine augmentations into groups and each head in LooC learns representations with respect to the groups instead of individual augmentations. We believe it is a valid future direction to build on our work.
>
> We provide full ImageNet experiments of MoCo and LooC++. Pre-trained on IN-1k, then fine-tuned (linear protocol) and benchmarked on iNat-1k, MoCo archives 47.8/74.3 top1/top5 accuracy, while LooC++ achieves 51.2/76.5 top1/top5.
>
> | Model      |   top1 |   top5 |
> | ----------- | ----------- | ----------- |
> | MoCo      |  47.8 |  74.3 |
> | LooC++ | 51.2 | 76.5 |
>
>
> **Q2. Table 4 and the different of MoCo++ and LooC++**
>
> **A2.** We would like to note that when interpreting the results in Table 2 to 5, the readers should keep Table 1 in mind as its broader context. Table 1 illustrates the catastrophic consequences of not separating the varying and invariant factors of an augmentation (in this case, rotation). It can be imagined that if we add “rotation classification” as one  downstream task in Table 4, MoCo++ will perform as poorly as it is in Table 1. The key of our work is to avoid what has happened in Table 1 and simultaneously boosts performance.
>
> LooC++ learns varying and invariant factors of given augmentations, which is one of the key contributions of our paper, whereas MoCo++ only learns invariances with any augmentation. They have embedding spaces of the same dimensions.

---

### Official Review · AnonReviewer2 · 2020-10-29
**Official Blind Review #2**

**Rating:** 8
**Confidence:** 4

**Review:**

Summary:
The authors observe that, while effective, contrastive learning unavoidably introduces some bias depending on the choice of augmentations the algorithm is made invariant to, and that deteriorates performance depending on the task. The authors corroborate this hypothesis with experiments with the MoCo baseline and proceed to propose a modification to the usual contrastive learning setup: learning a shared representation and multiple projection heads, each invariant to a different augmentation type. They empirically show the effectiveness of the proposed solution on several tasks, including few-shot learning and data corruption datasets.

Great:
* The structured approach to understanding which augmentations help and how to tackle the issue of choosing is addressing a significant issue in the contrastive learning literature. The authors thoroughly motivate their approach. While it's overall clear in the literature that contrastive learning is extremely successful at building generalizable representations, the choice of which augmentations to use is often arbitrary. The authors show empirically how making a model invariant to specific augmentations is  detrimental to some tasks (e.g. adding rotation invariance degrades 100-category ImageNet accuracy).
* Extensive evaluation on several different tasks, showing consistent improvement on all and highlighting the flexibility of the proposed approach. The authors show results on a coarse-grained task (ImageNet 100 and iNaturalist 2019), fine-grained (CUB-100 and VGG Flowers), an augmentation-specific task (ObjectNet, real-word objects with different views and rotations) and a robustness task (ImageNet-C).

Questions:
* The proposed method suggests that the best performance is achieved by choosing more and more augmentations. Do the authors believe there is a limit after which adding more augmentation heads makes the task impossible for contrastive learning, by asking the joint embedding space to account for too much flexibility?
* In practice, even when using the proposed method, one still has to choose a relatively small set of possible augmentations. In light of their work, do the authors have thoughts on how should someone make that choice?

Overall:
The paper is proposing a well empirically motivated modification to current contrastive learning methods, the methods and results are presented clearly, the experiments are extensively described. A clear accept.

---

> ### Author Response · Authors · 2020-11-19
> **Response from the Authors**
>
> We would like to thank the reviewer for the comments. Please find the following for our response.
>
> **Q1. Whether there is a limit for adding more augmentation?**
>
> **A1.** When the number of augmentations grows, it is inevitable that by adding augmentations in a brutal force means the learnt representation gets a bit saturated as there are too many varying and invariant factors to learn. A straight-forward hypothesis is to combine augmentations into groups and each head in LooC learns representations with respect to the groups instead of individual augmentations. We believe it is a valid future direction to build on our work.
>
> **Q2. How to choose candidate augmentations for self-supervised learning?**
>
> **A2.** Choosing augmentations is an important question to all self-supervised learning algorithms. Unfortunately, all of the current self-supervised learning frameworks that are based on instance discrimination and contrastive learning have the same issue. Our paper makes a critical step towards being less sensitive to a hand-crafted set of augmentations. We can simultaneously learn augmentation invariant/sensitive information and let the downstream task choose automatically. As for augmentations that have no benefits for any downstream task whatsoever, we believe that identifying these augmentations is orthogonal to our work.

---

### Official Review · AnonReviewer1 · 2020-10-31
**Tackling an important problem, not achieving the expected results**

**Rating:** 5
**Confidence:** 4

**Review:**

== Summary ==

The paper proposes a contrastive learning approach for self-supervised learning in which multiple heads are trained to be invariant to all but one type of data augmentation. The rationale is that different downstream tasks may require different types of invariances (e.g. we may want to be rotation invariant for pictures of flowers, but not for pictures of animals), and one does not know a-priori which kind of invariances will be required. After training multiple representation heads, one can later concatenate them or use the general embedding (the input to all the variant-specific heads) for the downstream task.

== Pros ==

- The authors try to tackle an important problem of self-supervised approaches: how does one decide which data augmentation strategies to use when the downstream task is not known in advance? This question has not been properly addressed in the literature, and can be of great important for the real application of self-supervised strategies beyond academic benchmarks.

- The method that the authors introduce scales well with the number of data augmentations used (linear), and avoid a combinatorial explosion that could arise.

- The paper is generally well written and the algorithm is explained quite clearly, and illustrations are used appropriately to help the reader understand the proposed approach (Figure 1 and Figure 2).

== Cons ==

- Once the self-supervised pre-training finishes, one has to decide whether to use the general embedding space in the downstream task, or a concatenation of the different variant-specific embeddings. However, from the results reported in Table 2, 3 and 4 it's not clear which approach is better, and this is downstream task-dependent. This is unfortunate since it basically introduces another hyperparameter to tune for each downstream task.

- Many experiments do not report any measure of variance or statistical significance, and do not follow an an "standard" setting. This makes really hard to tell whether the observed increase in accuracy is statistically significant or not. For instance, the authors use IN-100 and ON-13, two subsets of the ImageNet-1k dataset which this reviewer has never seen before, and thus the results are really hard to interpret. The only results that show some measure of variance (standard deviation) are with Flowers-101.
*Update after discussion*: Authors pointed out that IN-100 is in fact used in other works. During the discussion they also provided additional standard deviation measures for CUB-200. Although I believe that reporting standard deviation of multiple runs and/or confidence intervals for the results should be the standard practice, I acknowledge the effort made by the authors running additional experiments to accommodate this demand at least for some of the datasets that they use.

- When training using all (3) data augmentations, the results in Table 4 don't suggest that LoCo improves upon MoCo in any significant way. The same applies for LoCo++ vs. MoCo++. Some of the reported accuracies are indeed slightly higher than the baseline, but the differences are small. In addition, for the only dataset for which the authors report confidence intervals, these are greatly overlapping in most of the cases.

- The authors restricted their experiments to a ResNet-50. A few experiments showing that their approach works for other modern architectures would be appreciated (e.g. DenseNet, EfficientNet, Inception).
*Update after discussion*: The authors provided additional results using a ResNet-101. I appreciate the effort of the authors running these extra experiments.

- There are plenty works using contrastive losses for doing self-supervised learning, yet the authors decided to compare only against the MoCo baseline. Additional baselines would be also appreciated. This reviewer acknowledges that using MoCo would probably be sufficient if the results were significantly better.
*Update after discussion*: The authors argue that MoCov2 is the strongest baseline at the moment of submission, which is publicly available and can run in affordable resources. This is a perfectly valid point.

== Reasons for score ==

I believe that the authors aim to tackle a very important question in self-supervised learning. However, the proposed approach fails to deliver the expected results, according to the results shown in Table 4. Other tables show LoCo and LoCo++ shine on top of the baseline (MoCo), but this is only "constrained" settings. The truth is that Table 4 contains the best numbers for the baseline, thus these are the results that LoCo(++) should improve, but it is not clear that it does so. As mentioned before, the differences are too small in most of the cases, and no measure of statistical significance is reported for most datasets. Only with Flowers-102 some measure of variance is reported (it's not clear whether the +/- show standard deviation or confidence intervals), and the intervals largely overlap in most cases.

*Update after discussion*: The authors have addressed most of my concerns, although not always satisfactorily. They made a considerable effort running additional experiments to provide with additional standard deviations of the accuracy in CUB-200, as well as provided results achieved using a ResNet-101. They also clarified some of my concerns regarding the datasets used. In some situations, the benefit of the proposed approach versus already existing methods is not clear, but in others the experimental evaluation shows clears benefits. Given this, and the fact that the paper is well written and motivated, I am increasing my score.

---

> ### Author Response · Authors · 2020-11-19
> **Response from the Authors**
>
> We would like to thank the reviewer for the comments. Please find the following for our response.
>
> **Q1. “It’s not clear whether general embedding space (LooC) or concatenated subspaces (LooC++) is better”. It may “add another hyperparameter to tune”.**
>
> **A1.** We would like to note that from Table 2, 3, 4 it’s clear that LooC++ outperforms or is comparable with its counterpart LooC in most circumstances, when the two use the same set of augmentations, except for Flowers-102 dataset. We conjecture that it is because Flowers-102 is a few-shot dataset therefore LooC++’s higher dimensional representation tends to overfit.
>
> Even if it were true that whether to use LooC or LooC++ added another hyperparameter, it only added **one** experiment during fine-tuning. In comparison, in traditional methods, e.g., MoCo, suppose there are 3 augmentations. If each augmentation is binary (w/ or w/o the augmentation), we must conduct 2^3=8 experiments to determine the augmentations. If each augmentation has 5 scales from 0 to 1, we must conduct 5^3=125 experiments to determine the best combination of augmentations. It is significantly less efficient than LooC.
>
> **Q2. “Many experiments do not report any measure of variance or statistical significance, and do not follow an ‘standard’ setting (e.g., IN-100 and ON-13)”.**
>
> **A2.** We observed in our experiments that the variance of results is very low (+-0.2), which is in accordance with the open sourced MoCov2 (+- 0.1) [1], except for few shot settings. Please note that [1] only reports the variance in its github repo rather than the white paper.
>
> It’s universally accepted to omit variance **when data is sufficient**. Notable examples include ResNet [2] and DenseNet [3]. Throughout our research careers, deep learning with sufficient data is surprisingly reproducible with low variance. It is common, on the other hand, to report variances on few-shot settings for rigorous experiments, as small scale data poses higher variances.
>
> We didn’t create the splits of ImageNet-100 (IN-100) by ourselves, instead we followed the split as published work [4] and its popular re-implemented repo of multiple self-supervised algorithms. (we apologize for not including this detail in the draft). ObjectNet-13 (ON-13) is created by taking the overlapping classes of ObjectNet, which has ~100 classes as IN-1k, and IN-100, yielding 13 overlapping classes. It’s not an arbitrary choice and we believe they are standard settings.
>
> **Q3. “The results in Table 4 don't suggest that LoCo improves upon MoCo in any significant way”. The same applies for LoCo++ vs. MoCo++.**
>
> **A3.** We would like to note that when interpreting the results in Table 2 to 5, the readers should keep **Table 1** in mind as its broader context. Table 1 illustrates the catastrophic consequences of not separating the varying and invariant factors of an augmentation (in this case, rotation). It can be imagined that if we add “rotation classification” as one  downstream task in Table 4, MoCo++ will perform as poorly as in Table 1. The key of our work is to avoid what has happened in Table 1 and simultaneously boosts performance. We will include the discussion in the new draft.
>
> **Q4. The authors “restricted their experiments to a ResNet-50”.**
>
> **A4.** Due to time constraints we hereby provided ResNet-101 experiments for comparisons. We train the baseline MoCo, and LooC with color and rotation leave-one-out augmentation with ResNet-101 backbone on IN100, then finetune (linear protocol) and benchmark their performance on iNat1k.
>
> | Model  | top1  | top5 |
> |----|----| ----|
> |MoCo| 35.9 | 61.4 |
> |LooC| 43.9 | 69.4 |
>
> **Q5. “There are plenty of works using contrastive losses for doing self-supervised learning, yet the authors decided to compare only against the MoCo baseline”.**
>
> **A5.** At the time when the draft was submitted, MoCov2 was the strongest baseline which was 1) publicly available; 2) reproducible with normally affordable resources (MoCov2 can run on a single machine with 8 gpus, whereas SimCLRv2 and BYOL, are implemented on TPUs with dozens of cores).
>
> Furthermore, our core contribution is to study the invariance problem that is common in contrastive learning, so it should not matter much what the baseline is as long as the algorithms are based on instance discrimination and contrastive learning. Eventually we selected a simple baseline since it is always hard to illustrate and dissect when multiple new components are intertwined.
>
> [1] https://github.com/facebookresearch/moco
> [2] Deep Residual Learning for Image Recognition, He et al.
> [3] Densely Connected Convolutional Networks, Huang et al.
> [4] Contrastive Multiview Coding, Tian et al., https://github.com/HobbitLong/CMC/
> [5] Objectnet: A large-scale bias-controlled dataset for pushing the limits of object recognition models, Barbu et al.
> [6] Big Self-Supervised Models are Strong Semi-Supervised Learners, Chen et al.

---

> > ### Comment · AnonReviewer1 · 2020-11-24
> > **Thanks for the clarifications**
> >
> > I appreciate the clarifications. I will discuss the paper further with the other reviewers, but for now I will keep my score for the following reasons:
> >
> > - You argue that results from Tables 2 to 5 should be read keeping results from Table 1 in mind. Indeed, Table 1 shows the catastrophic effects "of not separating the varying and invariant factors of an augmentation", for an artificial task: predicting rotations. The question is how important is this for the real tasks at hand. It's still not clear to me from results in the other tables.
> > - You "observed in [your] experiments that the variance of results is very low (+-0.2)". If you did run the experiments multiple times and measured the variance, what's the reason to not include it in the paper? "It’s universally accepted to omit variance when data is sufficient", the question is whether the experiments reported in the paper fall in that scenario or not. iNat has hundreds of thousands of images, IN-100 has dozens of thousands as well, but CUB-200 has about 6k. Is this "sufficient"? Again, if you computed the variance of the results, I would suggest to add them in the paper to make the results more robust.

---

> > > ### Author Response · Authors · 2020-11-25
> > > **Further clarifications**
> > >
> > > We would like to thank the reviewer for the timely feedback.
> > >
> > > 1. We would like to note that the Table 1 is not simply an “artificial task”; it can be reasonably imagined that in a mobile app one feature is to distinguish whether an image is upside down before further processing. On the contrary, Table 1 represents the fundamental flaws of current contrastive learning paradigms from a higher perspective: every time an augmentation is added, to some extent the information w.r.t. the augmentation is lost. Until today contrastive learning frameworks released on the internet are still setting impressive records on ImageNet validation set, but we ought to ask the essential question that “are self-supervised contrastive learning spiraling towards a local minimum while ignoring the bigger picture”. The issue has to be addressed at some point. We believe and would like to highlight that our work is important in its unique contribution to the field, and that our key contribution should not be diminished as the request for a further one or two percentage improvement.
> > >
> > > 2. Per the reviewer’s request we have conducted multiple runs on CUB-200. Due to time constraint we have chosen our method LooC with color augmentation. We ran the experiment five times, the result is 40.1 +- 0.2 / 69.8 +- 0.25 top1/top5 vs 40.1 / 69.7 reported in the white paper. It proves that it is in fact sufficient and is aligned with our research experience. We don’t think the variance on ImageNet or iNaturalist will exceed it on CUB200.
> > >
> > > We hope that the reviewer would reconsider the recommendation as we do not believe that the questions raised should warrant rejection of the submission.

---

### Public Comment · ~Hankook_Lee1 · 2020-11-17
**Related work**

Dear Authors,

Thank you for the interesting paper. I want to draw your attention to our previous work [1] related to this work. Our work [1] aimed at improving fully-supervised learning using self-supervised transformations such as rotations. To this end, we assigned different labels for each self-supervised transformation, which leads to relaxing an invariance constraint with respect to the transformations.

While the idea of removing invariance properties w.r.t. transformations is similar to ours, I think the technique of augmenting embedding spaces is very interesting. Moreover, LooC can improve pre-training without labels while ours can improve fine-tuning with labels; thus, two frameworks seem to be complementary, and using both might provide additional gains in downstream tasks. Like your work, I also think that studying how to utilize the input transformations effectively would play a crucial role in representation learning.

Thank you very much.

[1] Hankook Lee et al., Self-supervised Label Augmentation via Input Transformations, ICML 2020, https://arxiv.org/abs/1910.05872

---

### Decision · Program_Chairs · 2021-01-07
**Final Decision**

**Decision:**

Accept (Poster)

**Comment:**

There was a predominantly positive feedback from the reviewers so I recommend acceptance of the paper. It is well-written and well-motivated tackling an important problem: That in self-supervised learning one might encode different invariances by default, even if some of these invariances are useful for downstream tasks (e.g. being rotation invariant may be detrimental to predicting if an image has the correct rotation on a phone). For this, they propose a simple, yet elegant approach and validate it on many downstream tasks. Given the recent interest in self-supervised learning, this appears to be a relevant and interesting paper for the ICLR community.